# Comparison between the Effect of Lidocaine or Ropivacaine Hematoma Block and of Different Timings of Administration on Post-Operative Pain in Dogs Undergoing Osteosynthesis of Long-Bone Fractures

**DOI:** 10.3390/ani13182858

**Published:** 2023-09-08

**Authors:** Irene Dimopoulou, Tilemachos Anagnostou, Ioannis Savvas, Panagiota Karamichali, Nikitas Prassinos

**Affiliations:** 1Anaesthesia, Analgesia, Emergency and Critical Care Unit, Companion Animal Clinic, School of Veterinary Medicine, Faculty of Health Sciences, Aristotle University, 54627 Thessaloniki, Greece; tanagnos@vet.auth.gr (T.A.); isavas@vet.auth.gr (I.S.); pkaramich@hotmail.com (P.K.); 2Surgery & Obstetrics Unit, Companion Animal Clinic, School of Veterinary Medicine, Faculty of Health Sciences, Aristotle University, 54627 Thessaloniki, Greece; ngreen@vet.auth.gr

**Keywords:** analgesia, dog, hematoma block, lidocaine, local anaesthetics, ropivacaine

## Abstract

**Simple Summary:**

Post-operative pain after fracture fixation in dogs can be severe. We already know that intra-operative bupivacaine hematoma block (HB, the infusion of local anaesthetic–bupivacaineto the fracture site during the operation) is effective in reducing post-operative pain in dogs undergoing long-bone fracture fixation (Dimopoulou et al. 2017). The aim of the present study was to compare the efficacy of other local anaesthetics, such as lidocaine or ropivacaine, for post-operative pain relief when administered via an HB in the same clinical scenario and also to investigate which is the best intra-operative time point to perform the HB in order to achieve better post-operative pain relief. For this aim, we infused two different local anaesthetics (lidocaine or ropivacaine) at the same surgical time point to compare their analgesic efficacy but also the same local anaesthetic (ropivacaine) at three different distinct surgical time points. Post-operative pain was estimated with the use of the University of Melbourne Pain Scale (a multimodal pain scale) and an algometer (a device that measures the mechanical pain threshold). The results of the present study indicate that the dogs that received a lidocaine HB experienced better post-operative pain relief than those who received a ropivacaine HB. Furthermore, the dogs that received the HB (ropivacaine) right before surgical closure had better post-operative pain relief.

**Abstract:**

Objective: We aimed to compare the efficacy of intra-operative lidocaine hematoma block (HB) to ropivacaine HB and to compare the efficacy of different timings of ropivacaine HB in controlling post-operative pain in dogs undergoing the osteosynthesis of long-bone fractures. Study Design: We conducted a randomized, blinded, prospective clinical study. Animals: Forty-eight dogs with long-bone fractures were included and were randomly allocated to four groups: lidocaine (L), ropivacaine (Rmid), ropivacaine pre- (Rpre) and ropivacaine post- (Rpost) groups. Methods: The dogs in group L (n = 14) and in group Rmid (n = 11) received a lidocaine or ropivacaine HB, respectively, after fracture reduction and before osteosynthesis material placement. Rpre dogs (n = 11) received ropivacaine HB before fracture reduction, and Rpost dogs (n = 12) received ropivacaine HB after osteosynthesis material placement. Eight post-operative pain assessments were performed using the University of Melbourne Pain Scale (UMPS) and an algometer. Rescue analgesia was administered based on UMPS scoring. For data analysis, the Shapiro–Wilk test of normality, chi-square, Student t test and Split Plot analysis were used. The level of significance was set at α = 0.05. Results: Rescue analgesia was administered to one dog in group L, one in group Rmid and one in group Rpost, with no significant differences detected. Compared to group Rmid, group L dogs exhibited significantly higher mean mechanical pain thresholds (*p* = 0.049) and lower mean UMPS scores (*p* = 0.001). Group Rpost dogs had statistically significantly higher mean pain thresholds compared to group Rmid (*p* = 0.009). Clinical Implications: When performed after fracture reduction and before osteosynthesis material placement, lidocaine HB seems to be more effective than ropivacaine HB in controlling post-operative pain in dogs undergoing osteosynthesis of long-bone fractures. The administration of ropivacaine HB after osteosynthesis material placement seems to be more effective than administration after fracture reduction and before osteosynthesis material placement or administration before fracture reduction in controlling post-operative pain in dogs undergoing osteosynthesis of long-bone fractures.

## 1. Introduction

Hematoma block (HB)is the injection of local anaesthetic directly into the fracture site [1], which, in human medicine, is considered a safe and effective alternative for a closed reduction in distal radius fractures without inferior pain relief compared to procedural sedation and analgesia in adults and paediatric patients [1,2,3,4,5,6,7,8,9].

The use of HB intra-operatively in order to reduce post-operative pain was first attempted in 2004 in children during femoral elastic nailing. In that study, bupivacaine 0.5% was injected into the fracture hematoma. Hematoma block was considered a quick, simple and effective method for post-operative pain relief. The block did not significantly lengthen the time of surgery, and it postponed the need for opioid administration for approximately five hours compared to the control group [10].

Comparative studies looking into lidocaine, levobupivacaine and ropivacaine have been published using several types of locoregional techniques and species [11,12,13,14]. In the veterinary literature, to the authors’ knowledge, there is only one study on dogs describing the intra-operative intra-fragmentary installation of bupivacaine or saline. This study concluded that the bupivacaine HB-modified technique can aid in post-operative pain relief in dogs that are submitted to long-bone osteosynthesis [15].

To the authors’ knowledge, no comparative study of local anaesthetics’ post-operative analgesic efficacy when used in HB techniques in dogs has been conducted. The aim of the present study was to compare the efficacy of intra-operative lidocaine HB to ropivacaine HB and also to compare the efficacy of different timings of intra-operative ropivacaine HB administration in controlling post-operative pain in dogs undergoing osteosynthesis of long-bone isolated fractures. The hypothesis of the study was that the intra-operative ropivacaine HB would provide better post-operative pain relief compared to lidocaine HB. A secondary hypothesis was that the ropivacaine HB performed before fracture reduction would provide better post-operative pain relief compared to the ropivacaine HB performed after fracture reduction and before osteosynthesis material placement or performed after osteosynthesis material placement.

## 2. Materials and Methods

This was a randomized, blinded, prospective clinical study. It was approved by the Institution’s Ethical Committee (82/6-7-2017). Informed consent was obtained from all owners whose dogs were included in the study. ARRIVE (Animal Research: Reporting of In Vivo Experiments) guidelines protocol was followed in all cases. Animals included in this study were dogs (1) with isolated diaphyseal long-bone fractures that underwent osteosynthesis surgery, (2) older than six months and (3) characterized as ASA (American Society of Anesthesiologists) physical status II–III cases. Dogs with intra-articular fractures or any other accompanying severe injuries were excluded due to potential confounding pain issues.

During the three and a half years of this research, 80 dogs were presented to the clinic for osteosynthesis of long-bone fractures. Of these, 32 were not included in the study, because they did not meet the inclusion criteria or met at least one of the exclusion criteria. The flowchart below (Figure 1) presents, in detail, the reasons why those dogs were excluded. Twelve animals were excluded because they had more than one fracture, ten because other sources of pain coexisted (multiple soft tissue injuries, thoracostomy tube placement), nine because they were younger than six months and one due to breed considerations (Boxer judged unsuitable to receive acepromazine premedication) (Figure 1).

After diagnosis, all dogs received carprofen (Rimadyl; Zoetis, CA, Canada) at a dose of 2 mg kg^−1^ intravenously (IV) twice daily until surgery. The dogs were fasted for 12 h before surgery and had free access to water up to one hour before induction of anaesthesia.

On the day of the scheduled surgery, the data that were recorded for all animals preoperatively were the following: (1) Scoring of soft tissue damage (1—mild: minimal soft tissue damage, superficial abrasion and/or contusion, simple or mild fracture pattern; 2—moderate: deep abrasion, localized skin and muscle contusion, moderate fracture pattern; 3—severe: extensive skin contusion or crushing, severe damage to underlying muscle, severe fracture pattern (modified from Tscherne and Oestern 1982)] [16]. (2) Affected bone(s) (humerus, radius, ulna, femur, tibia). (3) Location of fracture line (distal, middle or proximal diaphysis). (4) Number of bone fragments (simple or comminuted). (5) Presence or absence of fragment displacement. (6) Communication between the fracture site and the environment (open or closed). (7) Direction of the fracture line (transverse, oblique or spiral). (8) Method of fixation (plating, intramedullary pinning accompanied by orthopaedic wire, external osteosynthesis). Mechanical pain threshold measurements were also obtained from all animals using an algometer (Vetalgo Algometer; Bioser, BP 32025, F—13845, Vitrolles Cedex) (see below for details) before pre-anaesthetic medication was administered [17]. During pre-anaesthetic assessment, heart and respiratory rate were also recorded as well as “mental status” (as described in the UMPS (Firth & Haldane 1999)) in order to evaluate post-operative alterations for the UMPS scoring [18]. 

Pre-anaesthetic medication included acepromazine (Acepromazine; Alfasan, The Netherlands) 0.05 mg kg^−1^ IM and morphine (morphine sulfate; Famar SA, Thiva, Greece) 0.2 mg kg^−1^ IM. Shortly after induction of anaesthesia and during surgical preparations, carprofen (Rimadyl, Zoetis, CA, Canada) 4 mg kg^−1^ IV was also administered. Lactated Ringer’s (L-R Lactated Ringer’s Injection, Vioser, Greece) was administered intra-operatively at 5 mL kg^−1^ h^−1^. As an induction agent, propofol (Propofol MCT/LCT Fresenius; Fresenius Kabi Hellas, Greece) at an initial dose of 2 mg kg^−1^ IV was used, and additional doses of 1 mg kg^−1^ were infused if needed. The trachea was intubated with a cuffed endotracheal tube (KRUUSE PVC, KRUUSE A/S, Denmark) of appropriate size. Isoflurane (AErrane; Baxter Healthcare Ltd., Norfolk, UK) in oxygen was delivered through a rebreathing circle system for animals weighing 7 kg or more (oxygen flow rate 1.5 L min^−1^) or a Jackson-Rees’ modification of an Ayre’s T-piece (Veterinary Anesthesia Systems Inc., Phoenix, TX, USA) for animals weighing less than 7 kg (oxygen flow rate 2.5-times the minute ventilation taken as 200 mL kg^−1^ min^−1^).

The dial of the vaporizer was originally set at 2.5% but, within a few minutes, it was adjusted to provide the appropriate depth of surgical anaesthesia, based on clinical assessment (evaluation of eyeball position, palpebral reflex, pupil light reflex, reflex movement in response to nociception and mandibular muscle tone), on end-tidal isoflurane concentrations (Et_ISO_) and on parameters indicative of the function of the cardiovascular and respiratory systems. In particular, all animals included in the study were monitored with electrocardiography (lead ΙΙ), pulse oximetry, arterial blood pressure measurement (indirectly, oscilometric method) (Datex-Ohmeda S/5, GGE Healthcare Finland Oy, Finland), capnography, measurement of O_2_ and isoflurane concentrations in inspired and end-tidal gas and of tidal volume and airway pressure (Capnomac Ultima; Datex-Engstrom, Helsinki, Finland). Intra-operatively, all animals received fentanyl (Fentanyl; Janssen Pharmaceutica NV, Belgium) 0.1 μg/kg^−1^ min^−1^ IV as a constant rate infusion (CRI). Fentanyl administration was discontinued at the end of the surgery. Fentanyl administration could be increased if the animal intra-operatively showed clinical signs of pain (elevation of heart rate, respiratory rate or blood pressure in adequate depth of anaesthesia). In an attempt to avoid hypothermia during anaesthesia, heating pads were used in all cases.

Local anaesthetic (lidocaine or ropivacaine) was injected to the fracture site during surgery based on group allocation. All forty-eight dogs that met the inclusion criteria and none of the exclusion criteria were randomly allocated into four groups using a random numbers table according to the local anaesthetic injected into the fracture site and according to the timing of infusion. The two different local anaesthetics (lidocaine and ropivacaine) were injected at the same orthopaedic timing, in the middle of the procedure, after fracture reduction and before the placement of osteosynthesis materials in group L and group Rmid. The same local anaesthetic (ropivacaine) was infused at three different distinct surgical time points: (1) after the surgical access and the debridement of the fracture sites, but before the fracture reduction and before the placement of osteosynthesis materials (group Rpre); (2) after fracture reduction and before the placement of osteosynthesis materials (group Rmid), as already described; and (3) after the placement of osteosynthesis materials right before the closure of the surgical site (group Rpost).Group L dogs received lidocaine 2% (Xylozan 2%, DEMO S.A., Krioneri, Greece) at 2.6 mg kg^−1^ (0.13 mL kg^−1^) via injection to the fracture site, as accessed through the surgical approach. Group Rpre, Rmid and Rpost dogs received ropivacaine 0.75% (ropivacaine Kabi 7.5 mg mL^−1^) at 1 mg kg^−1^ (0.13 mL kg^−1^) also via injection to the fracture site, as accessed through the surgical approach. The infusion of the local anaesthetic was performed using a syringe with a 25 gauge 5/8-inch needle (the tip of the needle was guided to the fracture site extra-medullary, allowing for slight contact to the fracture line and also including, inevitably, the bone periosteum and the surrounding soft tissues in all cases.

Each animal had only one local anaesthetic injection. The injection syringes were prepared by one of the authors (T.A.), and the injections were performed by the chief surgeon (N.P.). The syringe to be used was made available to the chief surgeon at the beginning of surgery, and anaesthetists were kept unaware of the timing of administration. Thus, the anaesthetist (I.D.) recording pre-operative data, monitoring anaesthesia and evaluating pain pre- and post-operatively was always the same and was blinded to the treatment group. Manipulations of the bones were temporarily stopped for two minutes after the infusion to allow for absorption of local anaesthetics into the fracture site, but sham manipulations were performed by the surgeons during this period to keep anaesthetists blinded concerning the time of treatment. During sham manipulations, the surgeons used their surgical instruments, pretending that they were working on the fracture area but without touching it. They also performed some manipulations to the neighbour healthy area like wiping the blood.

Eight pain assessment was performed post-operatively using the UMPS and the algometer. The first assessment took place within thirty minutes after extubation followed by seven more assessments, 1, 2, 4, 6, 8, 20 and 32 h later.

The algometer that was used to measure the mechanical pain threshold as a response to pressure application (operation principle is based on the Von Frey filaments) consists of a load cell, a handle, a recording device and tips. The algometer’s measurement range is 0 to 5000 g. The device records the maximum pressure or weight applied to the tip until end of contact of the tip with the surface (skin). All algometer measurements were performed with minimal restraint and in a quiet environment to ensure that even mild reactions of dogs undergoing assessments would be perceived. The tip was put into contact with the skin with gradually increasing force applied to create noxious stimulation. Pressure was increased until specific kinds of reactions were noted on the part of the dog: withdrawal of the limb and/or escape attempts and/or vocalization. When such a reaction was noted, application of pressure was immediately stopped. On each pain evaluation, pain threshold measurements were performed on three sites: (a) on the skin covering the fracture line (near the incision line) (FL), (b) on a region of healthy skin (with no signs of inflammation or irritation) nearest to the incision line (NFL) and (c) on an area of the contralateral healthy limb corresponding to the fracture area of the affected limb (CHL). Three measurements were obtained from each site, and the mean of measurements was calculated and recorded [15].

The UMPS evaluates six categories of data or behaviours associated with the response to pain: physiologic data, response to palpation, activity, mental status, posture and vocalization. Each parameter is scored, and the total score ranges from 0 points (absence of pain) to 27 points (worst pain imaginable). If the UMPS score equalled or exceeded 15 points, rescue analgesia was administered to the dog, and the animal was excluded from further measurements. Rescue analgesia included fentanyl 3 μg kg^−1^ IV and morphine 0.3 mg kg^−1^ IM. Administration of morphine 0.1 mg kg^−1^ IM was repeated as needed to the dogs that received rescue analgesia [15]. 

Sedation was also scored simultaneously with each pain assessment using a simple descriptive scale (0 = alert, 1 = dog lightly sedated but responds readily to visual and auditory stimulation, 2 = dog sedated that responds only to intensive visual and auditory stimulation, 3 = dog heavily sedated unresponsive to intense visual and auditory stimulation) [15].

Post–operatively, all dogs received carprofen 2 mg kg^−1^ SC or PO every 12 h for 4 days. Restricted activity on a leash was recommended for all dogs until clinical fracture healing.

Comparison of results for mechanical pain thresholds and UMPS scores was performed between groups L and Rmid to investigate the efficacy of the two different local anaesthetics injected at the same time point (after fracture reduction and before the placement of osteosynthesis materials). Comparisons of results for mechanical pain thresholds and UMPS scores were performed between groups Rmid, Rpre and Rpost to investigate the efficacy of a local anaesthetic (ropivacaine) injected at 3 different time points (before fracture reduction, after fracture reduction and before the placement of osteosynthesis materials, after fracture stabilization) of the surgical procedure.

## 3. Statistics

The Shapiro-Wilk test was applied to determine whether data followed a normal distribution or not. The Pearson chi-square and Monte Carlo significance tests were used to determine the statistical significance. Split Plot ANOVA with two factors (treatment and time) and animals as a random factor was used to compare main effects. The level of statistical significance was set at α = 0.05. SPSS program package was used for the statistical analysis (IBM, SPSS, Statistics 27).

## 4. Results

The results are presented as mean ± standard deviation or median and range where appropriate.

Group L consisted of 14, group Rmid of 11, group Rpre of 11 and group Rpost of 12 dogs. The median age was 2.7 years (range 0.5–13). The mean weight was 18.1 ± 10.3 kg. The median administered dose of propofol per kg of body weight was 3.3 mg kg^−1^ (2–9.1). The mean duration of anaesthesia was 243 ± 59 min, and the mean duration of surgery was 174 ± 50 min. Statistically non-significant differences among groups were found regarding age (*p* = 0.167), weight (*p* = 0.748), degree of tissue damage (*p* = 0.565) and administered dose of propofol per kg of body weight (*p* = 0.523). There was a statistically significant difference between group L and group Rmid in anaesthetic (*p* = 0.021) and surgical (*p* = 0.008) time, with the duration of anaesthesia and surgery being longer in group Rmid (anaesthetic time 270 ± 39 min, surgical time 193 ± 47 min) than in group L (anaesthetic time 204 ± 73 min, surgical time 141 ± 47 min). Group Rmid differed statistically non-significantly from group Rpre and Rpost with regard to anaesthetic time (*p* = 0.656 and *p* = 0.320, respectively) and surgical time (*p* = 0.781 and *p* = 0.258, respectively).

Rescue analgesia was required for one dog in group L, one dog in group Rmid and one dog in group Rpost, with a statistically non-significant difference among the groups (*p* = 1). The overall (including all groups) percentage of dogs that received rescue analgesia was 6.3%.

The observed power of our analysis was 0.802 for the FL and 0.931 for the UMPS.

### 4.1. Comparison of the Efficacy of Lidocaine vs. Ropivacaine HB

The pre-anaesthetic (baseline) FL mechanical pain threshold was statistically non-significantly different between groups L and Rmid (*p* = 0.831). Differences between the two groups were also non-significant for NFL (*p* = 0.721) and CHL (*p* = 0.433).

The FL pain threshold post-operatively was statistically significantly higher in group L (1146 g ± 69) than in group Rmid (923 g ± 89) (*p* = 0.049). Group L had higher pain thresholds than group Rmidat almost all individual time points, except t20, but with the difference being non-statistically significant (Figure 2) (Table 1). The NFL pain threshold post-operatively differed non-significantly between groups L (734 g ± 64) and Rmid (840 g ± 83) (*p* = 0.314). The CHL pain threshold also differed post-operatively non-significantly between groups L (3828 g ± 83) and Rmid (3926 g ± 108) (*p* = 0.473).

The UMPS score in group L (4 ± 0.2) was statistically significantly lower than in group Rmid (6 ± 0.3) (*p* = 0.001). Group L had lower UMPS pain scores at all time points, with the difference being statistically significant at t_1_ (*p* = 0.026) and t_2_ (*p* = 0.003) (Figure 3) (Table 2).

The sedation score differed non-significantly between groups L (0.4 ± 0.04) and Rmid (0.3 ± 0.06) (*p* = 0.111). 

### 4.2. Comparison of the Efficacy of Ropivacaine HB Injected at Three Different Intra-Operative Time Points

The pre-anaesthetic (baseline) pain thresholds measured for FL were statistically non-significantly different among groups Rmid, Rpre and Rpost (*p* ≥ 0.655). The pre-anaesthetic (baseline) pain thresholds differed non-significantly for NFL (*p* ≥ 0.054) and for CHL (*p* ≥ 0.110).

The FL pain threshold obtained post-operatively was significantly higher in group Rpost (1226 g ± 73) compared to group Rmid (923 g ± 89) (*p* = 0.009). There was a non-significant difference between group Rpost and group Rpre (*p* = 0.393). Investigating individual time points, although group Rpost had higher FL pain thresholds than group Rmid at all time points until t_8_, a statistically significant difference was observed only at t_0_ between group Rpost (2414 g ± 207) and group Rmid (1195 g ± 217) (*p* = 0.000). Group Rpost also had higher pain thresholds than group R-pre at all time points until t_6_, but the difference between Rpost (2414 g ± 207) and Rpre (1460 g ± 217) was statistically significant only at t_0_ (*p* = 0.002) (Figure 4) (Table 3). 

The NFL post-operative pain threshold was higher in group Rpost (1223 g ± 68) than group Rmid (840 g ± 83) and group Rpre (978 g ± 76), with the difference being statistically significant (*p* = 0.000 and *p* = 0.017, respectively). Investigating individual time points, group Rpost had higher pain thresholds at all time points—except t_20_—compared to groups Rmid and Rpre. The difference between the Rpost and Rmid groups was statistically significant at t_0_ (*p* = 0.003) and at t_1_ (*p* = 0.001). The difference between the Rpost and Rpre groups was statistically significant only at t_0_ (*p* = 0.017).

The CHL pain threshold was 3926 g ± 108 in group Rmid, 3355 g ± 98 in group Rpre and 3386 ± 88 in group Rpost. Group Rmid had a statistically significantly higher CHL pain threshold than groups Rpre (*p* = 0.000) and Rpost (*p* = 0.000). Investigating individual time points, group Rmid had a statistically significantly higher pain threshold compared to group Rpre at t_8_ (*p* = 0.013) and at t_20_ (*p* = 0.013) and also significantly higher pain thresholds compared to Rpost at t_2_ (*p* = 0.029) and t_8_ (*p* = 0.007).

The UMPS score was significantly lower in group Rpost (5 ± 0.2) than in group Rmid (6 ± 0.3) (*p* = 0.06). The mean UMPS score differed non-significantly between groups Rpost and Rpre (5 ± 0.3) (*p* = 0.756). Investigating individual time points, at t_0_, group Rpost had a statistically significantly lower UMPS score than group Rmid (*p* = 0.005). At t_1_, group Rpre had a statistically significantly lower UMPS score compared to group Rmid (*p* = 0.004) (Figure 5) (Table 4).

The mean sedation score was 0.32 ± 0.06 in group Rmid, 0.4 ± 0.05 in group Rpre and 0.51 ± 0.04 in group Rpost. Group Rpost had a statistically significantly higher mean sedation score than group Rmid (*p* = 0.011). The mean sedation score differed non-significantly between groups Rpost and Rpre (*p* = 0.104) and between groups Rmid and Rpre (*p* = 0.576). Regarding individual time points, group Rpost had a significantly higher sedation score than group Rmid at t_0_ (*p* = 0.009) and t_2_ (*p* = 0.048). 

## 5. Discussion

The main finding of the present study, as shown both by algometer measurements and UMPS scores, is that lidocaine HB seems to be more effective than ropivacaine HB in controlling post-operative pain in dogs undergoing osteosynthesis of long-bone isolated fractures. Data also indicate that the administration of ropivacaine hematoma block after osteosynthesis material placement seems to be more effective than administration after fracture reduction and before osteosynthesis material placement or administration before fracture reduction in controlling post-operative pain in dogs undergoing osteosynthesis of long-bone isolated fractures.

Post-operative pain after long-bone fracture osteosynthesis can be severe [19,20,21,22]. This may lead to an extended use of opioids and prolonged hospitalization. Previous research regarding the intra-operative intra-fragmentary administration of bupivacaine showed significant efficacy for post-operative pain after fracture osteosynthesis in dogs [4]. In that study, no animals in the bupivacaine group required rescue opioid post-operatively, while in the control group, 67% of animals required opioids. Presumably, other local anaesthetics could also be potentially used in the same manner, in an attempt to reduce the use of opioids for post-operative pain relief.

The present study aimed to investigate the effectiveness of two other local anaesthetics, lidocaine and ropivacaine, used in the same manner. When compared to a previous relevant study by Dimopoulou et al. (2017) [15], the experimental design of the present study is identical, except for the use of different local anaesthetics for the HB. Exactly the same anaesthetic and analgesic protocol were used in order to avoid the use of a control group for ethical reasons. It has already been shown that animals given saline to the fracture site instead of a local anaesthetic (bupivacaine) are in significantly more intense pain. In the present study, the percentage of animals in all groups that needed rescue opioid after an HB was only 6.3%. This result is in agreement with the results of the previous study by Dimopoulou et al. (2017) [15] and highlights the potential role of intra-operative HB in the context of developing analgesic protocols with limited or no use of opioids post-operatively.

Opioids were used in the present study during pre-medication (morphine 0.2 mgkg^−1^) and intra-operatively (fentanyl CRI 0.1 μg kg^−1^ min^−1^) for reasons of sedation and analgesia. Some of the adverse effects of opioids are nausea, vomiting [23,24,25,26] and post-operative sedation [27], which are common both in human and veterinary medicine. Reduced post-operative use of opioids could limit the occurrence of such unwanted adverse effects. The clinical use of the HB technique could be a step towards opioid-free anaesthesia (OFA).

The results of the algometer mechanical pain threshold measurements and UMPS scoring both point to the direction that dogs in group L experienced less post-operative pain compared to group R-mid. This finding refutes our original hypothesis that ropivacaine would provide better post-operative pain relief, based on the prolonged duration of action of ropivacaine compared to lidocaine [28]. No information could be found in the literature regarding the comparison of the analgesic efficacy of lidocaine versus ropivacaine in an inflammatory environment, such as that of the fracture site (pH 6.69–6.89 in rats) [28]. Lidocaine could indeed act with greater analgesic efficacy in an acidic environment, since it has a lower pKa (7.9) than ropivacaine (8.1) [28]. Thus, in an acidic environment (fracture site), a greater proportion of a given dose of lidocaine would exist in the unionized and active form compared to a smaller proportion of ropivacaine [28,29,30,31,32].

The time when it would be most appropriate to administer the HB was also investigated in the present study by administering a ropivacaine HB at three different time points during surgery. The results of mechanical pain threshold measurements suggest better post-operative pain relief in the group Rpost compared to groups Rmid and Rpre, thus disproving the hypothesis that post-operative pain relief would be superior for group Rpre(based on the principle of pre-emptive analgesia). The UMPS results agree with the mechanical pain threshold results in showing better post-operative pain relief for group Rpost than for group Rmid but failed to indicate significant differences in pain relief between groups Rpost and Rpre. The significantly higher mean sedation score of group Rpost compared to group Rmid might be due to dogs in group Rpost being calmer because of better pain relief. It is likely that better pain relief in group Rpost was due to the block being implemented later on during the procedure compared to the other groups, thus extending the duration of action of ropivacaine into the immediate post-operative period compared to the other groups. The fact that the mechanical pain threshold was significantly higher in group Rpost than in group Rmid at time point t_0_ (first post-operative assessment, temporally close to the procedure and the block) supports this explanation. Moreover, the performance of the hematoma block after fracture fixation with osteosynthesis materials (group Rpost) may have helped in limiting the loss of anaesthetic out of the fracture site and toward surrounding tissues, which could have occurred in the Rpre and Rmid groups due to the more intense surgical manipulations that inevitably followed the administration of the block in those groups.

Chondrotoxicity is a concern with the intra-fragmentary injections of local anaesthetics. There are several in vitro studies demonstrating the toxic effects of local anaesthetics, especially bupivacaine, in articular cartilage cells [33,34,35,36,37,38]. According to a recent analytical systematic review, lidocaine, bupivacaine, ropivacaine, levobupivacaine and mepivacaine were reported to have dose- and time-dependent deleterious effects on chondrocytes. Ropivacaine at concentrations of 0.5% or less was found to be the least chondrotoxic, and bupivacaine 0.5% appeared to be the most chondrotoxic local anaesthetic [39]. 

There are also in vivo studies, both in humans and animals, demonstrating that local anaesthetics can have chondrotoxic effects when injected intra-articularly. In humans, several cases of chondrolysis have been reported in the glenohumeral joint after shoulder arthroscopy associated with the continuous infusion of bupivacaine, with or without epinephrine [40,41]. Similarly, in rabbits, intra-articular injections of bupivacaine into knee joints resulted in articular cartilage inflammation and synovial membrane changes [42].

In human medicine, while bupivacaine is considered to be the gold standard for intra-articular administration, ropivacaine has been used intra-articularly in several studies for post-operative pain relief. Ropivacaine seems a reasonable choice given the fact that it is less toxic than bupivacaine [43]. Although from a clinical point of view, chondrotoxicity seems to be a problem only after repeated intra-articular injections or continuous intra-articular infusions of local anaesthetics [36], it appears prudent to use less chondrotoxic local anaesthetics, like lidocaine or ropivacaine, when injecting into areas where bone healing occurs. In this respect, the use of lidocaine or ropivacaine for an intra-operative HB during osteosynthesis may be considered preferable.

Concerning the limitations of the present study, a statistically significant difference in anaesthetic and surgical time between groups L and Rmid was noted, with group L having shorter mean anaesthetic and surgical times. Since there are studies in the human medical literature that consider the increase in anaesthetic and surgical time as a predisposing factor for additional post-operative pain [44], one could consider that less intense pain in group L could be due to shorter surgical time and not to better analgesic efficacy of lidocaine when used for HBs. However, after careful inspection of the data, two cases of outliers were identified in group L, for which surgery lasted a short time compared to all other cases. Exclusion of those two cases resulted in non-significant differences in anaesthetic and surgical times between groups L and Rmid.

Group Rmid had significantly higher CHL pain thresholds than groups Rpre and Rpost. A possible explanation for this difference could be enhanced systemic absorption of ropivacaine in the Rmid group due to more intense surgical manipulations more closely temporally associated with the timing of the injection (after fracture stabilization and before the placement of osteosynthesis materials). Measurements of plasma concentrations of the local anaesthetics used for the HB at various time points would have been necessary to confirm or discard this assumption.

The omission of a control group that would have received a placebo HB intra-operatively could be considered another limitation of the study. As already stated, our recent work [15] showed that animals given saline to the fracture site instead of a local anaesthetic (bupivacaine) are in significantly more intense pain. Since exactly the same anaesthetic and analgesic protocol was used in the present and our previous study (except for the different local anaesthetics used in the HB) so that results could be directly comparable, it was decided that a control group (proven to suffer more intense post-operative pain) should not be included in the present study for ethical reasons. The question of whether an intra-operative HB with a local anaesthetic provides pain relief over placebo was answered in our previous study [15], and the aim of the present study was to compare between different local anaesthetics and different time points of administration.

## 6. Conclusions

In conclusion, lidocaine HB seems to be more effective than ropivacaine HB in controlling post-operative pain in dogs undergoing osteosynthesis of long-bone isolated fractures. Moreover, the administration of ropivacaine HB after osteosynthesis material placement seems to be more effective than earlier administration in controlling post-operative pain in the same clinical scenario.

## Figures and Tables

**Figure 1 animals-13-02858-f001:**
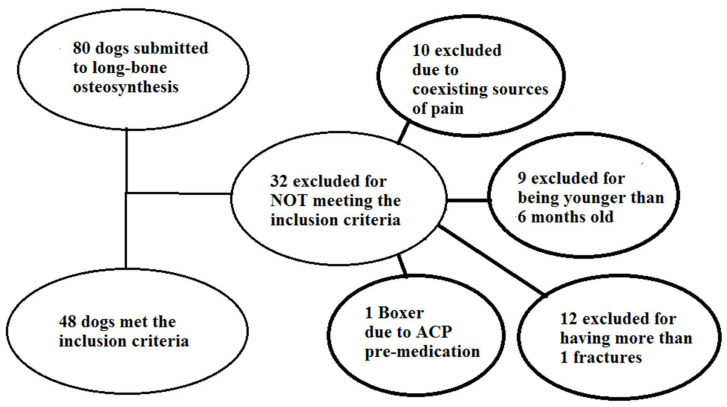
Flowchart for dogs submitted and reasons for exclusions.

**Figure 2 animals-13-02858-f002:**
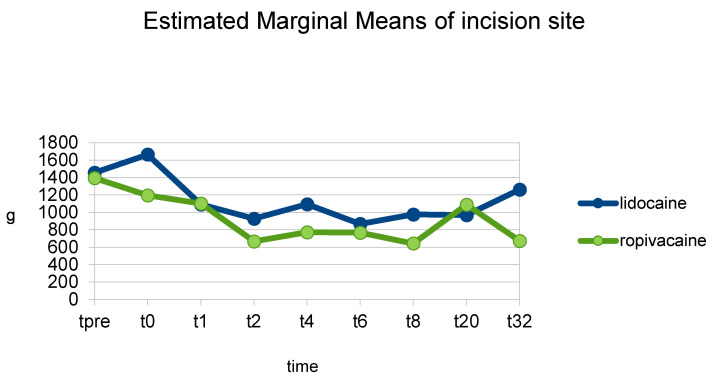
Mean pain threshold at the incision site at individual time points (t-pre to t_32_) in the groups of dogs that received either a ropivacaine (group Rmid) or a lidocaine (group L) hematoma block.

**Figure 3 animals-13-02858-f003:**
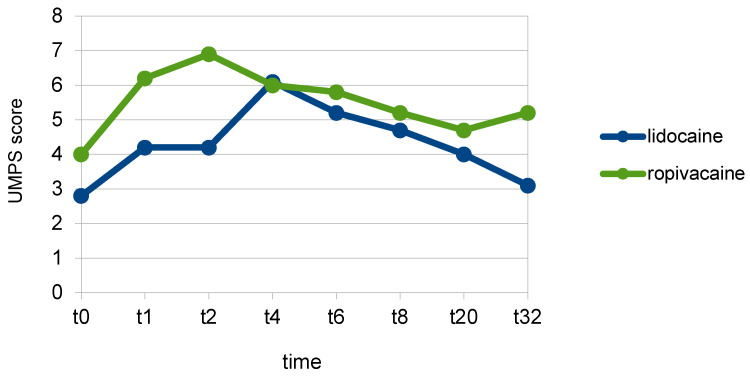
Mean UMPS scores at individual time points (t-pre to t_32_) in the groups of dogs that received either a ropivacaine (Group Rmid) or a lidocaine (group L) hematoma block.

**Figure 4 animals-13-02858-f004:**
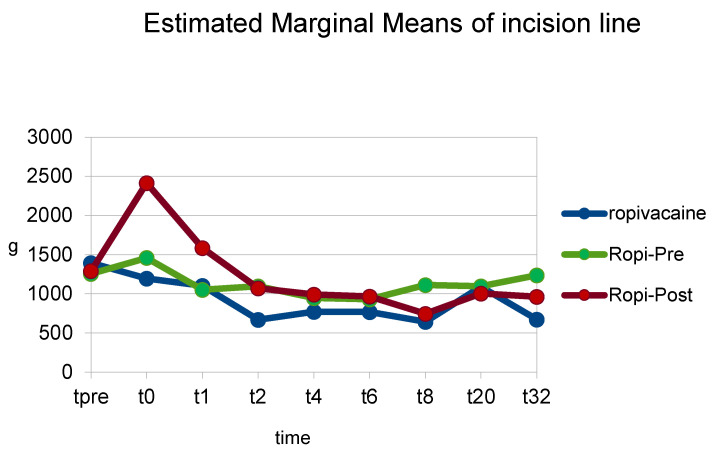
Mean pain thresholds at incision site at individual time points (t-pre to t_32_) in the groups of dogs that received a ropivacaine hematoma block before fracture reduction (group Rpre) or after fracture reduction and before the placement of osteosynthesis materials (group Rmid) or after the placement of osteosynthesis materials (group Rpost).

**Figure 5 animals-13-02858-f005:**
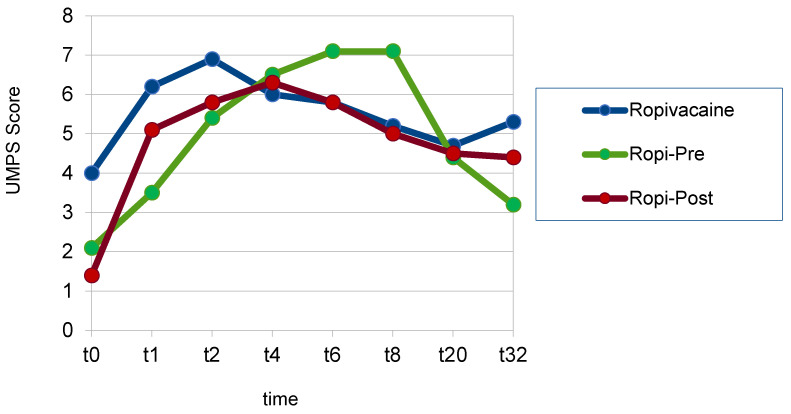
Mean UMPS scores at individual time points (t-pre to t_32_) in the groups of dogs that received a ropivacaine hematoma block before fracture reduction (group Rpre) or after fracture reduction and before the placement of osteosynthesis materials (group Rmid) or after the placement of osteosynthesis materials (group Rpost).

**Table 1 animals-13-02858-t001:** Mean pain threshold at the incision site at individual time points (t-pre to t_32_) in the groups of dogs that received either a ropivacaine (group R-mid) or a lidocaine (group L) hematoma block.

	Group L (g)	Group R (g)	Significance (*p*)
t-pre	1457	1395	0.831
t0	1665	1195	0.106
t1	1092	1102	0.973
t2	926	669	0.388
t4	1093	773	0.304
t6	869	768	0.752
t8	976	644	0.296
t20	970	1090	0.737
t_32_	1262	672	0.156

**Table 2 animals-13-02858-t002:** Mean UMPS scores at individual time points (t-pre to t_32_) in the groups of dogs that received either a ropivacaine (group R-mid) or a lidocaine (group L) hematoma block.

	Group L	Group R	Significance (*p*)
t_0_	2.8	4	0.168
t_1_	4.2	6.2	0.026
t_2_	4.2	6.9	0.003
t_4_	6.1	6	0.974
t_6_	5.2	5.8	0.554
t_8_	4.7	5.2	0.645
t_20_	4	4.7	0.497
t_32_	3.1	5.3	0.084

**Table 3 animals-13-02858-t003:** Mean pain thresholds at the incision line at individual time points (t-pre to t_32_) in the groups of dogs that received a ropivacaine hematoma block before fracture reduction (group R-pre) or after fracture reduction and before the placement of osteosynthesis materials (group R-mid) or after the placement of osteosynthesis materials (group R-post).

	Group R (g)	Group R-Pre (g)	Group R-Post (g)	Significance (*p*) R-Rpre	Significance (*p*) R-Rpost	Significance R-Pre—R-Post
tpre	1395	1258	1293	0.655	0.732	0.909
t_0_	1195	1460	2414	0.388	0.000	0.002
t_1_	1102	1053	1586	0.873	0.108	0.077
t_2_	669	1098	1072	0.176	0.195	0.931
t_4_	772	947	992	0.595	0.495	0.880
t_6_	768	932	965	0.017	0.541	0.913
t_8_	644	1113	746	0.153	0.756	0.243
t_20_	1090	1098	1004	0.982	0.814	0.779
t_32_	672	1236	962	0.262	0.495	0.548

**Table 4 animals-13-02858-t004:** Mean UMPS scores at individual time points (t-pre to t_32_) in the groups of dogs that received a ropivacaine hematoma block before fracture reduction (group R-pre) or after fracture reduction and before the placement of osteosynthesis materials (group R-mid) or after the placement of osteosynthesis materials (group R-post).

	Group R	Group R-Pre	Group R-Post	Significance (*p*) R-Rpre	Significance (*p*) R-Rpost	Significance R-Pre—R-Post
t_0_	4	2.1	1.4	0.041	0.005	0.459
t_1_	6.2	3.5	5.1	0.004	0.228	0.074
t_2_	6.9	5.4	5.8	0.103	0.242	0.606
t_4_	6	6.5	6.3	0.679	0.766	0.894
t_6_	5.8	7.1	5.8	0.204	0.988	0.168
t_8_	5.2	7.1	5	0.053	0.901	0.028
t_20_	4.7	4.4	4.5	0.796	0.799	0.991
t_32_	5.3	3.2	4.4	0.183	0.476	0.418

## Data Availability

The data presented in this study are available on request from the corresponding author. The data are not publicly available due to medical confidentiality reasons.

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
