# Peer review of "Comparison between the Effect of Lidocaine or Ropivacaine Hematoma Block and of Different Timings of Administration on Post-Operative Pain in Dogs Undergoing Osteosynthesis of Long-Bone Fractures"

_animals, 2023, doi:10.3390/ani13182858_

Round 1

Author Response

Thank you for reviewing our manuscript. We would be glad to hearing from you.

Reviewer 2 Report

Good morning.

I have read your paper with great interest and I congratulate you. I share a word file in which I have included some food for thought that I hope will help you increase the quality of the work.

In general I can tell you that in my opinion the idea of ​​the project is not bad, that some sections are well written and are satisfactory. Others are well written but are a bit poor in content.

The most critical section in my opinion is materials and methods, I believe there is a lot of room for improvement with little effort.

Regards

Author Response

(The authors gave the same response as above.)

Reviewer 3 Report

Dear Authors,

Thank you for submitting this original research. The technique described is valuable for any veterinary setting as does not required any equipment and is easy to perform. 

Please see below my comments:

Did the dogs receive any additional medications post-operatively except the carprofen? 

Line 139-141: In regards to the technique, it looks like that the anesthetic is splashed on the surgical site, especially after implants are in place (where when it is pre-implants it can be infiltrated in the actual hematoma), if so I would suggest to explain the procedure more clearly.

Line 122-136-138-142: please replace "Installed" with "instilled" or "infiltrated" 

Line 229-230: which time point was the difference detected? From the diagram it looks like t0 but should be specified like you did for the other differences. 

For all the diagrams it would be helpful to identify the differences on the graph with a symbol. 

Author Response

Thank you so much for your kind words and the reviewing of our manuscript. We are looking forward to hearing from you.
